# Latent-TGF-β has a domain swapped architecture

Mingliang Jin [1,4], Robert I. Seed[2,4], Tiffany Shing[2], Li Wang [2], Junrui Li[3], Yifan Cheng [1,3,5] ✉ & Stephen L. Nishimura [2,5] ✉

The multifunctional cytokine TGF-β is a dimeric protein produced within a latent complex (L-TGF-β). Latency is maintained by disulfide linked homo-dimeric prodomains forming a ring encircling the non-covalently bound mature TGF-β homodimer. This configuration sterically inhibits mature TGF-β from binding to its receptors. For TGF-β to be activated and bind to its receptors it must either be released, or if not released, overcome steric hinderance within the latent complex. Integrin binding to L-TGF-β results in activation with or without release of TGF-β by deforming the ring through different yet incompletely understood mechanisms. The domain architecture of L-TGF-β, which is not clearly defined, is a gap in mechanistic understanding of L-TGF-β activation. Here we fill this critical gap-in-knowledge by definitive experimental evidence demonstrating a domain-swapped architecture of L-TGF-β.

TGF-β is an essential multifunctional cytokine with diverse functions in morphogenesis, extracellular matrix (ECM), and immune homeostasis[1]. Understanding the regulation of TGF-β function is paramount to dissect the roles of TGF-β in disease and to facilitate targeting for therapeutic benefit. TGF-β is expressed as a latent complex (L-TGF-β), in which the ring-shaped latency-associated peptide (LAP) encircles mature TGF-β and sterically hinders TGF-β from engaging with its receptors (TGF-βR1 and TGF-βR2), as revealed by crystal structures (Fig. 1a, b)[2,3]. L-TGF-β is further stabilized by linking LAP covalently to a type 1 transmembrane protein, GARP; this asymmetric complex is tethered by the GARP transmembrane domain to immune cell surfaces (Fig. 1a)[2]. For TGF-β to function, a process of activation must take place to sufficiently deform the LAP latent ring so that mature TGF-β is exposed to its receptors in the same cell with or without release for autocrine signaling, or released from the LAP ring to diffuse and bind to receptors on distant cells for paracrine signaling[4,5].

Two integrins, αvβ6 and αvβ8, account for the physiologic activation of TGF-β1 in vivo, since their combined loss of function results in early lethal tissue inflammation similar to *tgfb1* knock-out mice[6–9]. Each

of these integrins bind with relatively high affinity to the integrin RGD recognition motif located on the arm domain of L-TGF-β1, but each are reported to support TGF-β activation by distinct mechanisms, which may be due to their structural differences[6,10–13]. The αvβ6 integrin has a wide conformational ensemble and can assume the extended-closed and open poses typical of other integrins, which have β cytoplasmic domains that bind the actin cytoskeleton. Together, these features are thought to provide a pathway for tensile force transduction from intracellular cytoskeleton to deform L-TGF-β sufficient for release of mature TGF-β[10] (Fig. 1c). In contrast, the αvβ8 integrin only assumes one pose, extended-closed, typical of other integrins in a low or intermediate activation state, and the β8-cytoplasmic domain does not bind to the actin cytoskeleton[6,12]. These features suggest that αvβ8 mediates TGF-β activation without cytoskeletal force transduction.

In our earlier studies, we used cryogenic electron microscopy (cryo-EM) and cell-based assays to support an alternative model for integrin-mediated TGF-β activation that does not require release of TGF-β: the inherent flexibility (i.e., conformational entropy) of L-TGF-β is redistributed from the integrin binding site to the contralateral side of the LAP ring, which induces sufficient flexibility and deformation,

[1]Department of Biochemistry and Biophysics, University of California San Francisco, San Francisco, CA, USA. [2]Department of Pathology, University of California San Francisco, San Francisco, CA, USA. [3]Howard Hughes Medical Institute, University of California San Francisco, San Francisco, CA, USA. [4]These authors contributed equally: Mingliang Jin, Robert I. Seed. [5]These authors jointly supervised this work: Yifan Cheng and Stephen L. Nishimura. ✉e-mail: Yifan.Cheng@ucsf.edu; Stephen.Nishimura@ucsf.edu

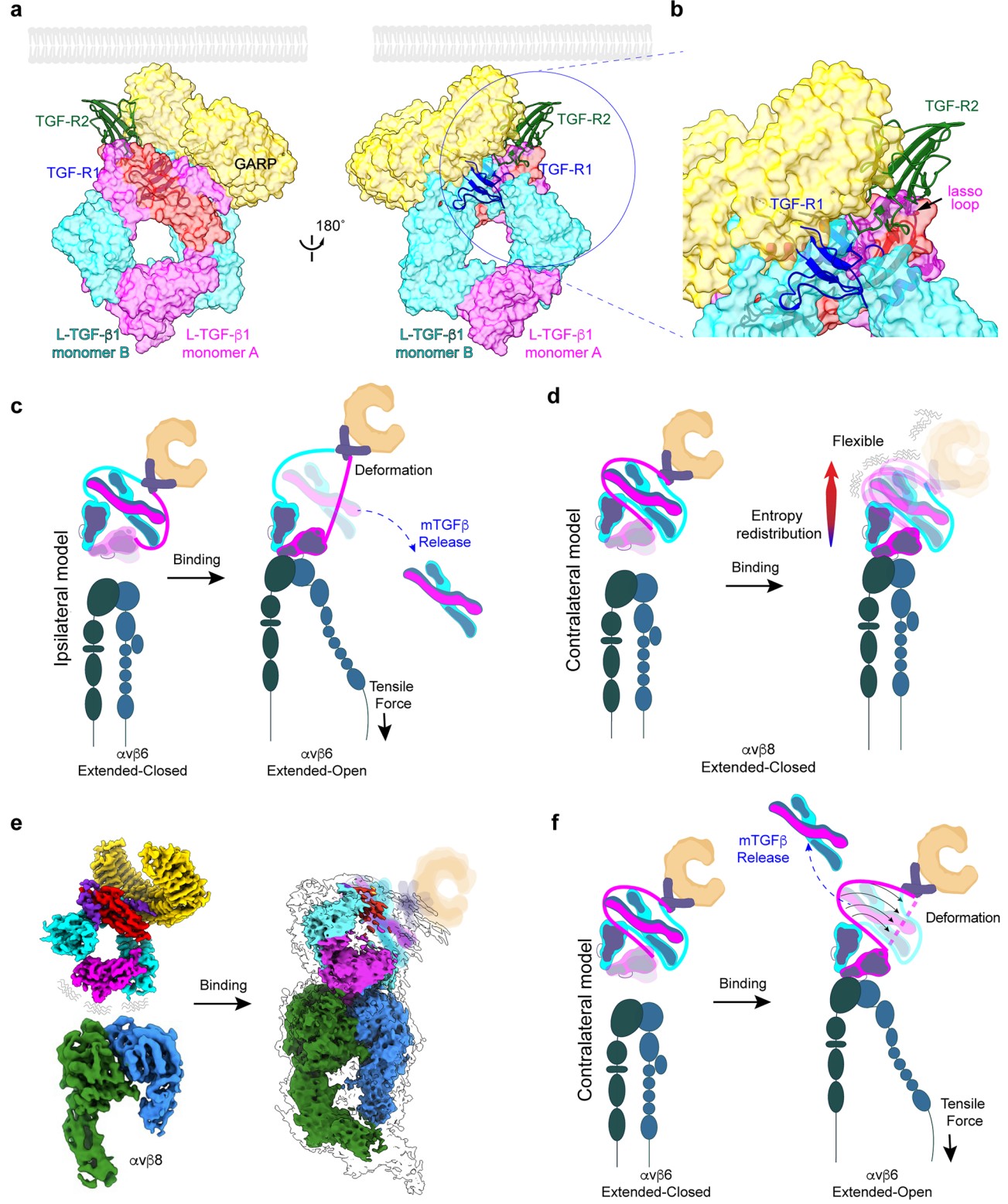

allowing mature TGF-β to overcome steric hindrance and bind to its receptors without release for signaling[4,5] (Fig. 1d, e). We then created mice with a loss-of-function mutation in the *tgfb1* furin cleavage site, resulting in mature TGF-β being covalently linked with LAP. These mice survived, maintained immune cell TGF-β signaling, and were rescued from the lethal early tissue inflammation seen in *tgfb1* knock-out mice, demonstrating that release of TGF-β1 from the latent complex is not required for its function[5].

Multidomain proteins, such as L-TGF-β, are folded into discrete structural units and assembled together with a defined overall three-dimensional (3D) domain architecture. These structural units (i.e., domains) are often connected through linkers, which are key to determining protein function. Thus, understanding domain architecture is critical for mechanistic understanding and therapeutic targeting. Interdomain linkers are often flexible, or even intrinsically disordered, and thus are typically poorly or not resolved by any

**Fig. 1 | Proposed mechanisms of integrin-mediated L-TGF-β activation.**
**a**, **b** Structural details of L-TGF-β1 latency. Superimposition of atomic models of L-TGF-β1/GARP (8VSC) with mature TGF-β1 (mTGF-β) in complex with TGF-βR1 (blue) and TGF-βR2 (green, 3KFD) reveals steric hindrance preventing binding of receptors to mature TGF-β by the lasso loop of the LAP. Two views are rotated 180°, with indicated enlarged view (circled region) in (**b**). GARP (gold), L-TGF-β1 monomer A (pink) or B (cyan), mTGF-β1$_A$ (salmon). Positioning of GARP and receptors relative to the membrane (grey) approximated based on predicted linkers lengths. **c** Intracellular cytoskeletal tensile force model of αvβ6-mediated L-TGF-β activation requiring the αvβ6 extended-open pose[10] derived from MD simulation based on ipsilateral architecture (not domain-swapped) without GARP. Coloring convention for monomer A/B is the same as panel **a**, with αv (green), β-subunit blue). **d** Model of αvβ8-mediated L-TGF-β activation[5]. Left: αvβ8 only in extended-closed pose

before, left, and after, right, binding to L-TGF-β. Note L-TGF-β is in contralateral configuration (domain-swapped) in proposed entropy redistribution model of TGF-β activation without release independent of intracellular cytoskeletal tensile force. Coloring as above. **e** Structures supporting entropy redistribution model[5]. Left: Cryo-EM density maps of unbound αvβ8 and L-TGF-β1. Right: selected class of αvβ8/L-TGF-β1/GARP complex, where densities of GARP and most of mature TGF-β with increased flexibility after αvβ8 binding. Coloring as above. **f** Proposed tensile force-induced TGF-β activation with L-TGF-β domain swapped. Left: unbound αvβ6, extended-closed, and L-TGF-β domain-swapped. Right: actin cytoskeleton tensile force transduced through extended-open β6 leg to contralateral L-TGF-β monomer. Coloring as above. Integrin cartoons created in BioRender. Nishimura, S. (2025) https://BioRender.com/29k4zvb.

commonly used structural technique. In such cases, the domain architecture cannot be determined with certainty, which is the case for TGF-β family members. Thus, determination of the domain architecture fills a gap in the structural understanding of the mechanism of TGF-β activation.

Of particular importance is the domain architectural arrangement in L-TGF-β. In all available crystal structures of L-TGF-β, there are two linkers that lack clearly defined densities (Supplementary Fig. 1a). One links the mature TGF-β to the LAP, and the other linker is within the LAP linking the arm domain with the straitjacket domain. An early study of L-TGF-β1 with a mutated furin cleavage site provided evidence that the arm domain in one monomer is linked to the contralateral mature TGF-β1 domain (i.e., swapped)[14]. In the same study, it also indicated that the linker between the straitjacket domain and arm domain remains undefined, with two possible ways to link the straitjacket and arm domains, resulting with two possible domain architectures where the straitjacket/lasso loop is either on the ipsilateral or contralateral RGD containing arm domain, both have been used in modeling integrin mediated L-TGF-β activation (Fig. 1c, d)[2,3,5,10,14,15]. AlphaFold prediction also gives both possible non-swapped ipsilateral and swapped contralateral architectures of the straitjacket/lasso (Fig. 2a). A better understanding of how integrin-mediated mechanical force or redistribution of conformational entropy is transduced across the LAP ring will require the correct assignment of the domain architecture.

In this work, we present a simple experimental design that definitively assigns the domain-swapped contralateral architecture to L-TGF-β, without the need for high-resolution structure determination. We further demonstrate that, without tensile force from the cytoskeleton, binding of the αvβ6 ectodomain also induces TGF-β signaling without releasing mature TGF-β. These findings provide key structural and cell biologic insights and that can be applied to better understand how integrin binding mediates L-TGF-β activation with or without release of mature TGF-β.

## Results
### Determination of the domain-swapped architecture of L-TGF-β
We recently determined the cryo-EM structure of L-TGF-β1/GARP, and noted a weak density linking the arm domain to the contralateral straitjacket/lasso domain (Supplementary Fig. 1b left)[5], contradicting the ipsilateral architecture presumed in the literature (TGFB2: 8FXS/8FXV[5,16], TGFB1: 8REW, 7Y1R[15], 5VQF[14], 5FFO[10], 5VQP[14], 8UDZ[17], 6GFF[2], 8C7H[18]) (Fig. 2a, left). However, this density is insufficiently robust to definitively define the domain architecture. Considering the flexible nature of this linker loop, further efforts to define the architecture definitively by pursuing high-resolution structure will likely be futile. We thus designed an alternative approach, in which the architecture can be unambiguously identified without the need for a high-resolution structure.

We generated two versions of a L-TGF-β1 expression plasmid (Fig. 2b), one with the intact RGD binding site and its TGF-β1 lasso loop (lasso1) replaced by the equivalent one from L-TGF-β3 (lasso3), and the

other has the intact lasso1 but contains a RGE mutation in its integrin binding motif. Both plasmids contain the R249A furin cleavage site mutation so that mature TGF-β1 remains covalently bound within the latent complex. We transfected these two plasmids in equal quantity together with GARP (Fig. 2c). The protein products, either in ipsilateral or contralateral architectures, consist of only three versions of the L-TGF-β1 dimer each covalently linked with a single GARP (Fig. 2c). One version of the dimer has a mixture of two mutant monomers, a heterodimer, one with a RGE motif and other with a lasso3 loop. This heterodimer has the RGD motif and lasso3 either on the same side of the LAP ring in ipsilateral or on opposite sides in the contralateral architecture (Fig. 2c left, DL3-EL1). The second is a homodimer with an intact RGD motif and a lasso3 in both monomers (Fig. 2c, middle, DL3-DL3). The third, also a homodimer, has both integrin binding motifs mutated to RGE, but both with an intact lasso1 (Fig. 2c, right, EL1-EL1). After mixing these three mutant forms of L-TGF-β1/GARP with integrin αvβ8 and an antibody 28G11, which recognizes L-TGF-β1 lasso1 but not lasso3, only the heterodimer (DL3-EL1) can bind one αvβ8 and one 28G11 at the same time. Examining whether αvβ8 and 28G11 are bound to the same side (only possible in the contralateral architecture) or opposite side (only possible in the ispilateral architecture) of the multicomponent αvβ8/L-TGF-β/GARP/28G11 complex will thus determine if L-TGF-β1 has an ipsilateral or contralateral domain architecture (Fig. 2c, Supplementary Fig. 1c, d).

To enrich the population of heterodimer that binds one integrin and one 28G11, we performed size exclusion chromatography (SEC) of the αvβ8 ectodomain with the mixture of L-TGF-β1/GARP mutants (Fig. 2d, Supplementary Fig. 2a–c). Peak C8 contains a higher proportion of heterodimer (DL3-EL1) that can only bind one αvβ8 (C8, Fig. 2d), and thus was used to incubate with 28G11 (Fig. 2e), which only recognizes the lasso1 (Fig. 2f, Supplementary Fig. 2d–g), and subjected to single particle cryo-EM analysis (Supplementary Fig. 3a–c).

We collected a modest single-particle cryo-EM dataset with only 1324 movies. Indeed, 2D class averages and 3D reconstruction of the complex, even at a modest nominal resolution of ~7 Å, clearly show that 28G11 only binds to the lasso on the same side as bound integrin. Docking the atomic models of L-TGF-β1/GARP and a generic Fab into this density map confirms this assignment (Fig. 2g, h, and Supplementary Fig. 3a–d). Together, our experimental design with a clear logic provides a concise and straightforward result demonstrating the domain swapped contralateral architecture (Fig. 2h, and Supplementary Fig. 3a–d).

### The domain swapped architecture of L-TGF-β supports activation without release by the integrin αvβ6
The domain swapped architecture of L-TGF-β has mechanistic implications for TGF-β activation. Based on our hypothetical model of autocrine TGF-β activation without release, any integrin that sufficiently stabilizes the TGF-β arm domain could result in entropy redistribution to the contralateral straitjacket/lasso sufficient to expose mature TGF-β to its receptors without release (Fig. 1d). The integrin αvβ6 also binds to and activates L-TGF-β but has not been

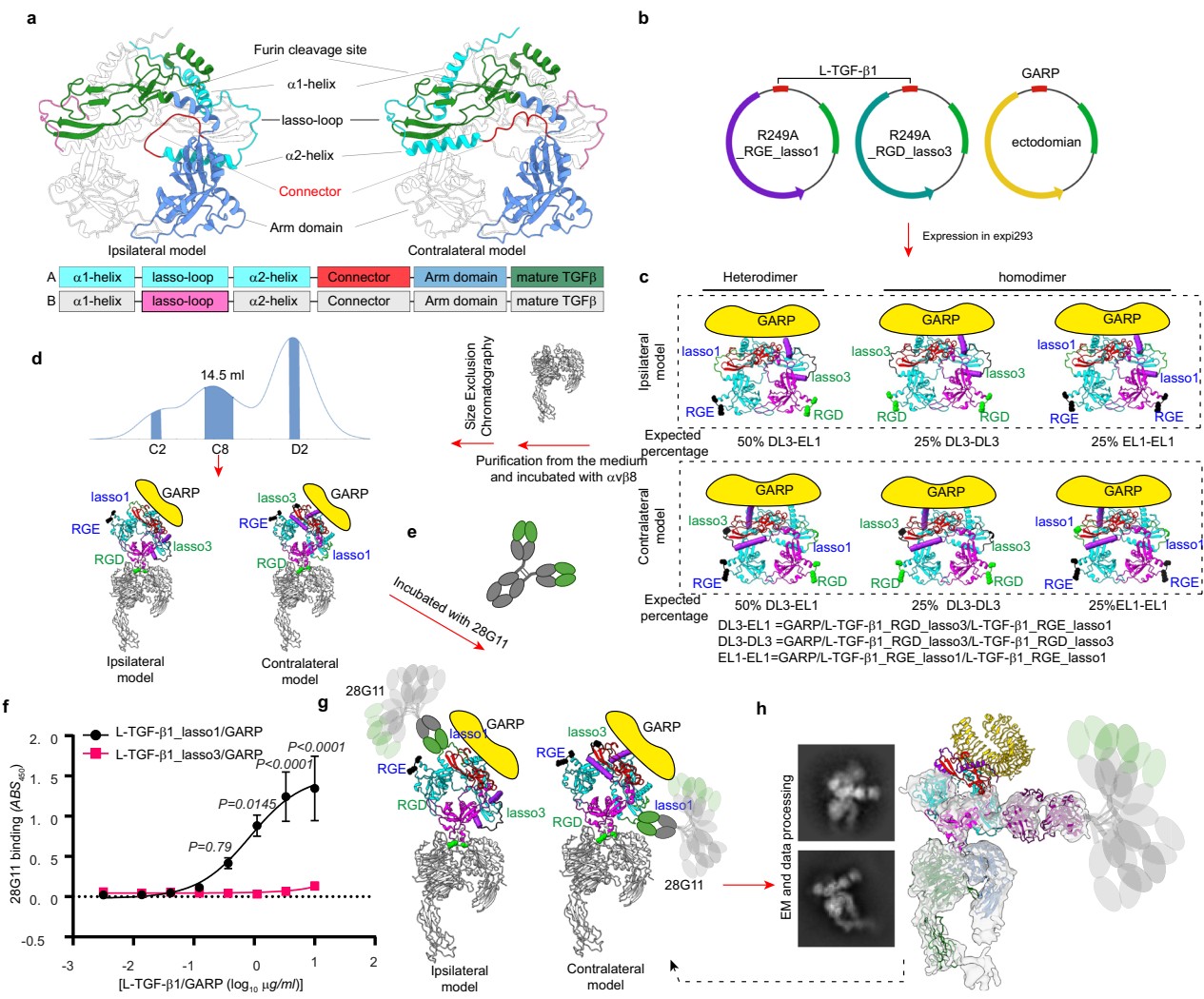

**Fig. 2 | Experimental design and determination of the domain architecture of L-TGF-β1/GARP. a** Ribbon diagrams of two possible architectures of L-TGF-β1 predicted by AlphaFold, in which the straitjacket domain of each monomer is connected ipsilaterally (left) or contralaterally (right) with its arm domain. Domain arrangement is illustrated under the ribbon diagrams. Color scheme in the ribbon diagram and domain arrangements are the same. **b** Plasmid constructs of L-TGF-β1_R249A_RGE_lasso1 (left), L-TGF-β1_R249A_RGD_lasso3 (middle), and GARP (right). **c** Anticipated expression products and their proportions from the 1:1:1 transfection of Expi293 cells. Nomenclature of the expected products is shown (heterodimer DL3-EL1 = GARP/L-TGF-β1_RGD_lasso3/L-TGF-β1_RGE_lasso1; homodimer DL3-DL3 = GARP/L-TGF-β1_RGD_lasso3/L-TGF-β1_RGD_lasso3; homodimer EL1-EL1 = GARP/L-TGF-β1_RGE_lasso1/L-TGF-β1_RGE_lasso1). Ribbon diagrams of L-TGF-β1 in ipsilateral (upper) or contralateral (bottom) architectures are predicted by AlphaFold. **d** SEC profile (middle) of purified mutant L-TGF-β1/GARP incubated with αvβ8 ectodomain (left). The shaded peak contains L-TGF-β1/GARP bound with a single αvβ8 with two possible architectures (bottom). **e** Incubation of monoclonal antibody clone 28G11 with the SEC-purified complex. **f** ELISA confirms that 28G11 only binds lasso1 but not lasso3. Data is presented as mean of absorbance$_{450}$ ($ABS_{450}$) ±SEM ($N = 4$ biologic replicates), $P$-values were generated by two way ANOVA with repeated measures followed by Šídák's multiple comparisons test. Source data are provided as a Source Data file. **g** The only possible models resulting from binding of 28G11 to mutant heterodimeric L-TGF-β1(DL3-EL1)/GARP in ipsilateral (left) or contralateral (right) architectures. The flexible regions of bound 28G11 IgG are indicated by cartoons. **h** Representative 2D class averages and 3D reconstruction with atomic models of a generic Fab and contralateral L-TGF-β1/GARP docked. GARP is partially seen in 2D class averages, but not in the 3D reconstruction. Part of mature TGF-β1 and remaining of IgG are illustrated but not seen in the density map.

reported to support TGF-β activation without release. Rather, unlike αvβ8, αvβ6 is hypothesized to require actin-cytoskeletal force-induced release of mature TGF-β from L-TGF-β[10]. We hypothesize that αvβ6 binding, even without such tensile force, stabilizes the arm domain and redistributes entropy sufficiently to support autocrine TGF-β activation without release. To test this hypothesis, we allow TMLC TGF-β reporter cells that co-express GARP and L-TGF-β1 with a mutated furin cleavage site (R249A) where mature TGF-β1 is covalently bound to LAP, to bind to immobilized αvβ6 ectodomain and then quantify autocrine TGF-β activation. This same assay was used in our previous study to demonstrate that αvβ8 activates TGF-β without release[4]. In this assay format, TMLC TGF-β reporter cells expressing GARP with similar surface levels of either wild-type L-TGF-β1 or L-TGF-β1 (R249A) (Fig. 3a),

support nearly identical amounts of αvβ6-mediated TGF-β activation (Fig. 3b). This result confirms that αvβ6, similar to αvβ8, is capable of mediating TGF-β activation without release in the absence of actin-cytoskeletal force.

αvβ6 can assume the extended-open pose and thus has a wider conformational range compared to αvβ8 which is limited to extended-closed[10,12,13]. This wide conformational ensemble of αvβ6 suggests that flexibility may play a role in tuning its function and ability to activate TGF-β without tensile actin cytoskeletal force. Our previous work suggests that conformationally flexible regions in integrins may serve as entropic reservoirs that can be manipulated by stabilization to alter the amount of entropy available for redistribution to L-TGF-β1/GARP[5]. Thus, we tested whether altering the amount of constraint of αvβ6,

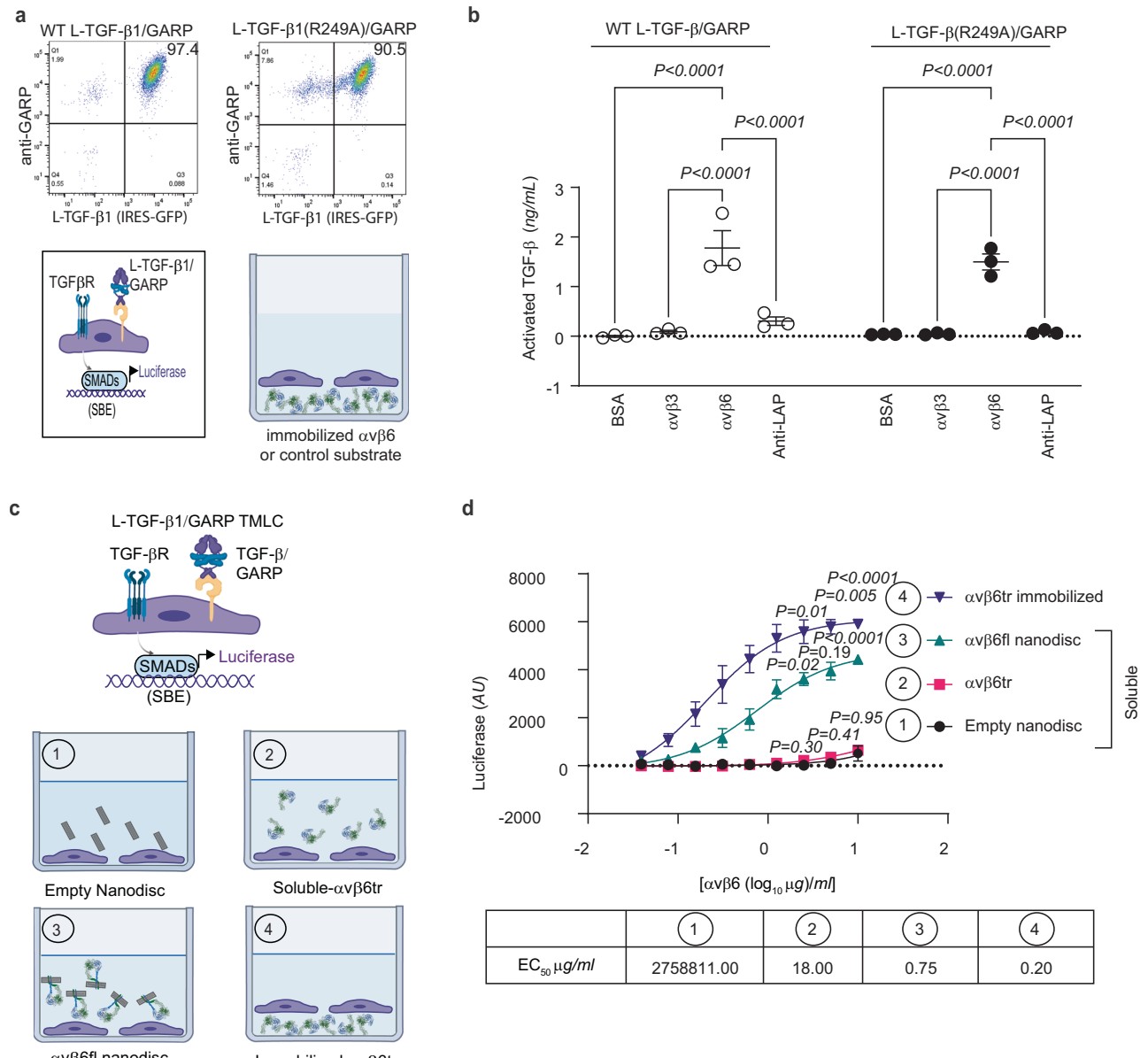

**Fig. 3 | A domain swapped architecture of L-TGF-β1 supports αvβ6-mediated autocrine TGF-β signaling without release. a** Cell-based system to measure autocrine TGF-β activation without release. Two lines are developed either expressing WT L-TGF-β1, or a furin cleavage site mutant L-TGF-β1 (R249A), where mature TGF-β1 remains covalently linked to LAP. Shown are scatterplots showing equal surface expression of both forms of L-TGF-β1/GARP in their respective lines. Below, a cartoon shows L-TGF-β1/GARP expressed on the cell surface of TMLC TGF-β reporter cells and applied to wells coated with immobilized αvβ6 ectodomain. **b** αvβ6 supports autocrine TGF-β signaling without release. TMLC cells either expressing L-TGF-β1/GARP or L-TGF-β1 (R249A)/GARP were allowed to bind to the immobilized αvβ6 ectodomain, or control substrates (all substrates coated at 1 μg/ ml), and amount of TGF-β activation determined by normalization to a recombinant TGF-β standard curve, as described[4]. The control substrates used are αvβ3 as a low-affinity TGF-β binding integrin, a polyclonal antibody to LAP as a high-affinity non-integrin L-TGF-β binder, or BSA, which does not bind to L-TGF-β. Data shown as

mean ± SEM, *P*-values were generated by one way ANOVA with Šídák's multiple comparisons test (*N* = 3 biologic replicates). Source data are provided as a Source Data file. **c** Cartoon of TMLC assay to measure effect of constraint on αvβ6 on TGF-β activation. Assays use L-TGF-β1/GARP TMLC either cultured with 1, empty nanodisc; 2, soluble αvβ6 ectodomain; 3, αvβ6 full-length in nanodisc; 4, immobilized αvβ6 ectodomain. **d** Constraint correlates with αvβ6-mediated TGF-β1 activation. Soluble αvβ6 ectodomain, full-length αvβ6 in nanodisc, or empty nanodisc as a control, were added to individual wells of TMLC L-TGF-β1/GARP cells, and TGF-β1 activation compared to TMLC L-TGF-β1/GARP cells incubated with immobilized αvβ6 ectodomain. The concentration of αvβ6 is indicated. TGF-β1 activation is indicated by luciferase activity in arbitrary units (*AU*). Data shown as mean ± SEM, *P*-values were generated by one-way ANOVA with Šídák's multiple comparisons test (*N* = 3 biologic replicates). Source data are provided as a Source Data file. Cartoons in (**a**, **c**) created in BioRender. Nishimura, S. (2025) https://BioRender.com/29k4zvb.

ranging from none to global stabilization, would affect TGF-β activation.

We compared TGF-ββ activation mediated by the αvβ6 ectodomain with no constraint (i.e. soluble) to maximal stabilization by global immobilization (Fig. 3c). TGF-β activation was inefficient and barely

detectable without constraint (soluble ectodomain) compared to immobilized αvβ6, which showed dramatic activation (Fig. 3d). To test whether constraining the transmembrane regions in a lipid membrane was sufficient to restore αvβ6-mediated TGF-β activation, we inserted full-length αvβ6 into lipid nanodiscs (ND). Indeed, TGF-β activation

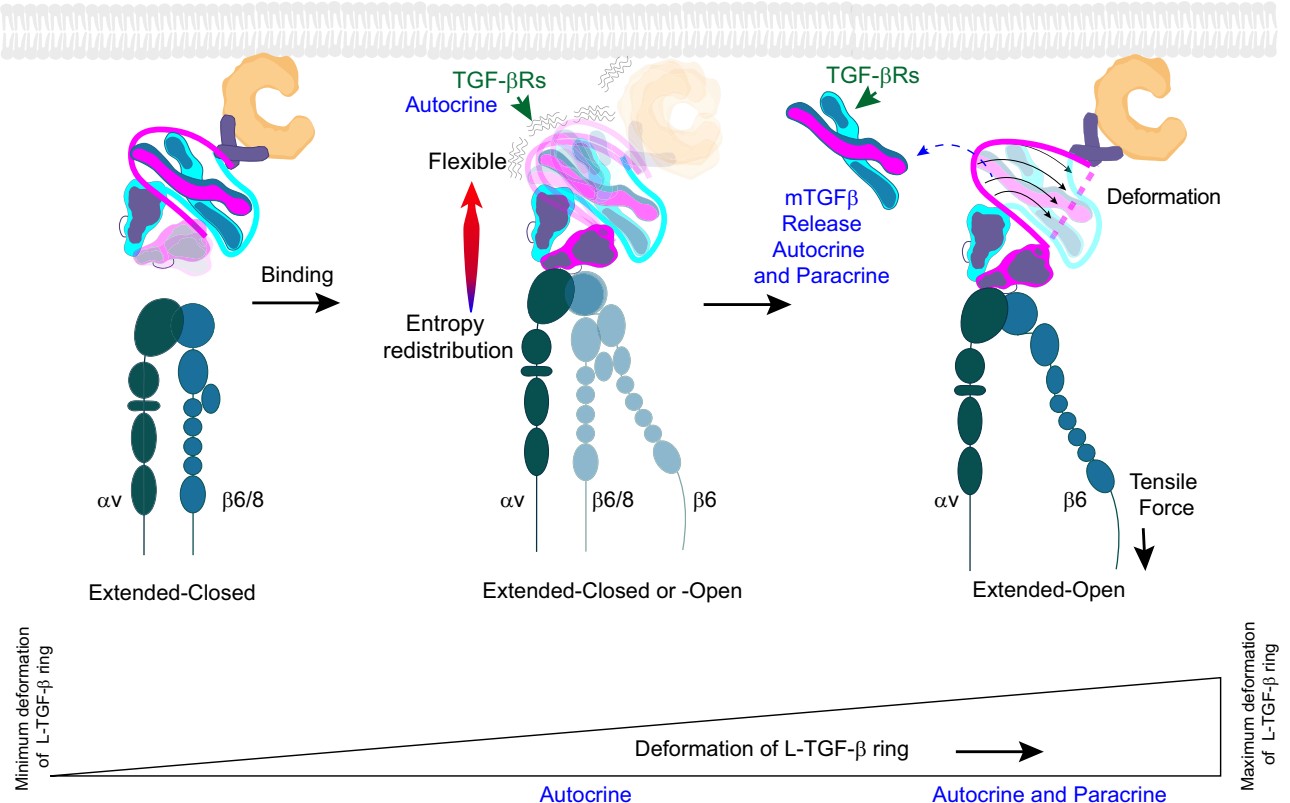

**Fig. 4 | Proposed consensus model of integrin-mediated TGF-β activation.** Proposed model of continuous deformation of L-TGF-β upon αvβ6 binding, based on L-TGF-β contralateral domain architecture, from intact latent ring with minimum deformation (left) towards fully deformed ring allowing mature TGF-β to be released (right). Increased deformation of L-TGF-β correlates with increased exposure of mature TGF-β to its receptors, from unreleased for autocrine signaling to the released form for paracrine signaling. Integrin cartoons are created in BioRender. Nishimura, S. (2025) https://BioRender.com/29k4zvb.

mediated by αvβ6-ND was significantly increased compared to the soluble ectodomain (Fig. 3d). Taken together, these data are consistent with our previous structural and cell-based data with αvβ8 and suggest that entropy redistribution is a shared mechanism of TGF-β activation between these two integrins[5]. Based on these results, we hypothesize that the domain swapped architecture of L-TGF-β is an essential structural feature providing a direct pathway for entropy redistribution (or tensile force transduction).

**Does the domain swapped architecture extend to other TGF-β isoforms and superfamily members?**
To address whether the domain-swapped architecture of L-TGF-β applies to other L-TGF-β isoforms, we examined the linker regions of L-TGF-β2 and L-TGF-β3 that are five residues longer than the L-TGF-β1 linker (Supplementary Fig. 3e). AlphaFold predicts that they both form α-helices and connect a domain-swapped architecture (Supplementary Fig. 3e). Interestingly, the cryo-EM structure of GARP/L-TGF-β3) contains a weak density in the region of the linker in L-TGF-β3/GARP[5] consistent with the domain-swapped architecture (Supplementary Fig. 1b right). Furthermore, examination of the amino acid sequence of the L-TGF-β3 (and L-TGF-β2) linkers reveals a central cysteine that predicts a disulfide bond with the other monomer in the contralateral architecture (Supplementary Fig. 3e). Taken together, these data suggest that L-TGF-β3 and L-TGF-β2, have domain-swapped architectures.

Does the domain-swapped architecture extend to other TGF-β superfamily members? Amongst other TGF-β superfamily members, the domain-swapped architecture was clearly seen in the crystal structure of Activin A, and was also assigned to myostatin, although a connector density was not seen (Supplementary Fig. 3f)[19,20]. We conclude that the domain-swapped architecture may be common within the TGF-β superfamily.

## Discussion
Prior to our study, structures of TGF-β family members have had an arbitrary assignment of the straitjacket/lasso domains ipsilateral to the arm domain. Using our approach, we determine with certainty that the straitjacket/lasso domain is not ipsilateral, rather it is contralateral to the arm domain. Taken together with previous studies[14], these data provide the final domain architectural details of L-TGF-β required to annotate existing TGF-β1 structures. This straightforward methodology can potentially be applied to determine the domain architecture of other related TGF-β superfamily members, which have prodomains that non-covalently link to their growth factors[21].

The importance of determining the contralateral domain architecture of L-TGF-β is that it provides a critical structural detail to assist in understanding the mechanism of integrin-mediated TGF-β activation, which remains incompletely understood. For instance, with the correct domain assignment molecular dynamic (MD) simulations can be performed to examine both force transduction and entropy redistribution. Such MD simulations have already been performed using an ipsilateral domain architecture[10]. These MD simulations applied retrograde force through the β6 cytoplasmic domain and found that the ipsilateral monomer underwent unfolding, prior to the contralateral monomer, which led to the release of mature TGF-β (Fig. 1c)[10]. In an ipsilateral architecture, these results are somewhat intuitive given that force should be effectively transduced through protein domains that are directly connected. Using the same logic, MD simulations of force-

induced αvβ6-mediated TGF-β activation with a domain-swapped architecture may lead to preferential unfolding of the contralateral straitjacket/lasso domain (Fig. 1f).

The correct assignment of the domain-swapped architecture sheds light on how the arm domain interacts with the contralateral straitjacket/lasso domain. Because we identified a density in the linker region between the arm and contralateral straitjacket domain of the L-TGF-β1/GARP cryo-EM map, the interdomain linker is at least partially constrained and not completely disordered. The core sequence of the L-TGF-β1 connector (PEPEPE) is highly conserved between species. Prolines create a linker more rigid than an intrinsically disordered linker, suggesting that the L-TGF-β1 linker functions in both spatial domain orientation and interdomain communication[22]. We hypothesize that the L-TGF-β1 arm domain connected through the proline-rich linker to the contralateral straitjacket/lasso domain produces a domain configuration permissive for both integrin-mediated force transduction and entropy redistribution to overcome steric hindrance and expose mature TGF-β to TGF-βRs. Exactly how such binding occurs remains to be determined since no structures of these complexes yet exist. Our finding that integrin αvβ6, like αvβ8, can support activation of TGF-β without release in the absence of actin cytoskeletal force supports the generalizability of our structure-based model of integrin-mediated L-TGF-β activation with a contralateral domain architecture[4,5].

Furthermore, the contralateral domain architecture appears to be generalizable to L-TGF-β3 and L-TGF-β2 isoforms, implying similar mechanisms of latency and activation. However, their interdomain linkers differ structurally from L-TGF-β1. The L-TGF-β3 and L-TGF-β2 linkers are each highly conserved between species and are flanked by a 4 amino N-terminal extension predicted by AlphaFold to extend the α2-helix, and lack the prolines found in the TGF-β1 linker (Supplementary Fig. 3e). L-TGF-β2 has a Cys residue at the C-terminus of its α2-helix extension and forms a disulfide bond with the other monomer[16]. Similarly, L-TGF-β3 has a Cys residue at the same position, presumably also forming a disulfide bond between two monomers with a domain swapped architecture (Supplementary Fig. 3e). These differences in the α2-helix and linker sequences suggest differences in interdomain communication, which may explain increased overall flexibility and lability (i.e., partial latency) of TGF-β2 and -β3 compared to TGF-β1, as well as a lowered threshold for integrin-mediated activation, either via force transduction and/or entropy redistribution[5,16,23].

Taken together, our current study with previous work[5,10,16] leads us to hypothesize that TGF-β activation is a continuum of exposure of mature TGF-β to TGF-βRs occurring from non-released and non-exposed (fully latent), non-released but partially exposed for autocrine signaling, towards fully released for both autocrine and paracrine signaling (Fig. 4). We speculate that in the contralateral domain architecture, cytoskeletal derived tensile force (for αvβ6) and/or conformational entropy (both αvβ6 and αvβ8) is transmitted through the interdomain linker leading to progressive deformation of LAP to effectively overcome steric hinderance and support TGF-β activation with or without release.

## Methods

### Recombinant protein expression

L-TGF-β1_RGD_Lasso3 (where the A31-L44 in lasso1 loop was swapped with T31-V42 from the L-TGF-β3 lasso3 loop) has been previously described in ref. 5. To produce L-TGF-β1 RGD_lasso3/L-TGF-β1 RGE_lasso1/GARP, Expi293 cells were transiently transfected with equal amounts of human L-TGF-β1 RGD_lasso3, L-TGF-β1 RGE_lasso1, and Strep-His-GARP plasmids as in Fig. 1c.

### Protein production

The secreted ectodomains of αvβ8 or αvβ6 or full-length αvβ6 integrins were produced by transfecting ExpiCHO cells with integrin

constructs following the previous protocol[4]. After 5 days of growth, cells were centrifuged, and the supernatant was filtered through a 0.2 μm PES membrane (Millipore). Protein was purified from supernatant via affinity chromatography using a Protein G column crosslinked with the 8B8 antibody, which binds to αv integrin[24]. Elution was achieved with 100 mM glycine (pH 2.5), followed by buffer adjustment and size exclusion chromatography (Superose 6 Increase 10/300 GL, GE Healthcare) in 20 mM Tris-HCl pH 7.4, 150 mM NaCl, 1 mM CaCl₂, and 1 mM MgCl₂. Full-length αvβ6 integrin was purified and inserted into lipid nanodiscs (ND), similarly as previously described with modifications[5]. Transfected cells were harvested 3 days post-transfection. Cells were solubilized by rotation at 4 °C using solubilization buffer for 3 h (20 mM HEPES, pH 8.0, 150 mM NaCl, 1 mM CaCl₂, 1 mM MgCl2, 10 mM DDM, 2 mM CHS, and 2% OG, 1x Protease Inhibitor Cocktail, EDTA-Free). Supernatant containing proteins was collected by centrifuged at 4000 g followed by ultra-speed centrifuge at 45,000 rpm. Protein purification is carried out by affinity chromatography using a column packed with Protein G crosslinked by antibody 3G9, which binds to αvβ6 integrin. Bound full-length αvβ6 is eluded from beads by washing the column with elution buffer (100 mM glycine at pH 2.5, 0.03% DDM). Flow through is immediately adjusted with 2 M Tris-HCl pH 8.0, followed by size exclusion chromatography (Superose 6 Increase 10/300 GL, GE Healthcare) in 20 mM Tris-HCl pH 7.4, 150 mM NaCl, 0.03% DDM, 1 mM CaCl₂, and 1 mM MgCl₂. αvβ6 in nanodisc (ND) was made by adding at a ratio of αvβ8fl: MSP-2N2: lipid equals to 1:4:200 in 4 °C for 3 hrs, biobeads were added to remove the residual lipids over night by gentle rotation. αvβ6-ND was collected and further purified by size exclusion chromatography (Superose 6 Increase 10/300 GL, GE Healthcare) in 20 mM Tris-HCl pH 7.4, 150 mM NaCl, 1 mM CaCl₂and 1 mM MgCl₂, the pooled and concentrated protein was subjected to SDS-PAGE, each protein size was identified to be correct.

The fractions corresponding to the mutant αvβ8/L-TGF-β1/GARP complex were eluted, collected into a Eppendorf tube, loaded for further SDS-PAGE and stained with Coomassie blue stain solution. The main 3 peak fractions were measured by Mass photometry performed with a Refeyn OneMP (Refeyn Ltd.). Each sample in TBS buffer with 1 mM CaCl₂, 1 mM MgCl₂ of 16 μl was pipetted into the reaction chambers. Calibration was carried out by BSA apoferritin and ADH. Each sample was diluted to 0.1 mg/ml 1 μl of each sample was added to a 15 μl TBS with 1 mM CaCl₂ and 1 mM MgCl₂ buffer already pipetted into the reaction chamber. Image analysis was performed and analyzed by the software provided by Refeyn Ltd. with the default settings provided by the manufacturer.

To produce secreted mutant L-TGF-β1/GARP, Expi293 cells were transiently transfected with three 3 plasmids: L-TGF- β1_R249A_RGE_-lasso1, L-TGF- β1_R249A_RGD_lasso3, and GARP ectodomain tagged with a Strep-His tag. The supernatant was collected by centrifuging the cell culture which grew for 5 days, and then filtered through a 0.2 μm PES membrane. Protein purification was done using Ni-NTA agarose, followed by washing with a buffer containing 0.6 M NaCl, 0.01 M Tris (pH 8.0), and elution with 250 mM imidazole in TBS. The eluted protein was applied to a Strep-tactin agarose column and washed with TBS. To remove the tag, HRV-3C protease was added, and the mixture was incubated overnight at 4 °C. Finally, the protein was concentrated to about 1 mg/ml in a TBS buffer using centrifugal concentrators.

Mutant L-TGF-β1/GARP and αvβ8 were first incubated at room temperature for 30 min, subjected to size exclusion chromatography and, correct peaks were pooled and concentrated to 0.31 mg/ml.

28G11 in IgG form purchased from Biolegend (San Diego, CA) was used without further purification.

### Cryo-EM

Purified mutant αvβ8/L-TGF-β1/GARP were incubated with 28G11 (1 mg/ml) at room temperature for 30 min at a molar ration of 1:1, the

final protein complex concentration is 0.37 mg/ml. For cryo-EM grid preparation, 3 μl of the complex was deposited onto Quantifoil 100 holey carbon films Au 300 mesh R 1.2/1/3, grids were glow-discharged for 30 s at 15 mA prior to sample application and freezing. The complexes were frozen using a FEI Vitrobot Mark IV using 1 s blot time. All grids were frozen with 100% humidity at 22 °C and plunge-frozen in liquid ethane cooled by liquid nitrogen.

The data set was collected on a Thermo Fisher 200 KeV Glacios equipped with a GATAN K3 direct detector camera. 1,324 movies were collected at a nominal magnification of 69,000x, the defocus range was set to be between −1.1 and −2.2 μm. The detector pixel size was 0.576 Å and the dose was 63 e⁻/Å². 

The data processing of αvβ8/L-TGF-β1/GARP/28G11 was carried out with CryoSPARC[25], with workflow shown in Supplementary Fig. 3a. The nominal resolution is estimated from the gold standard FSC = 0.143 criterion. Final reconstruction and directional FSC (cFSCs in CryoSPARC) show clear sign of anisotropic resolution, indicating that the dataset suffers from preferred orientation. An optical transfer function is calculated from the dFSC and used to deconvolute the final map[26]. The deconvoluted map is then low-pass filtered to 8 Å for atomic model fitting. Atomic model of αvβ8 and L-TGF-β1, both are taken from PDB 8VSD, a generic Fab are docked into the corresponding density of the deconvoluted map to calculate cross-correlation by using UCSF Chimera[27]. The location of 28G11 on L-TGF-β1 matches the previous cryo-EM structure of L-TGF-β1/GARP/28G11[18].

### X-ray map density calculation
The structure factor of L-TGF-β1 (PDB: 5VQF) was obtained from PDB and converted to mrc file which can be recognized by UCSF Chimera in COOT.

### Correlation between maps
Models of αvβ8/mutant L-TGF-β1/GARP/28G11 were fitted into the αvβ8/mutant L-TGF-β1/GARP/28G11 map. The correlation was calculated by Chimera *fit in map* module, in which αvβ8, L-TGF-β1, and 28G11, each was used to simulate an 8 Å map, and fit to the main map, correlation was shown in Supplementary Fig. 2b.

### AlphaFold prediction
The predictions of human L-TGF-β1 dimers, were performed using two identical TGF-β chains without signal peptide or templates by AlphaFold2 (https://colab.research.google.com/github/deepmind/alphafold/blob/main/notebooks/AlphaFold.ipynb)[28].

### Sequence alignments
Multiple protein sequence alignments for L-TGF-β were generated using Clustal Omega[29].

### Antibody binding assay
ELISA plates were coated with serial dilutions of recombinant TGF-β1/GARP or recombinant TGF-β1_lasso3/GARP (10 μg/ml) in PBS for 1 h at RT. Wells were then washed in PBS and blocked (5% BSA) in PBS for 1 h at RT. 28G11, 7B11, or isotype control antibody was added (1 μg/ml) in PBS for 1 h at RT. After washing in PBS tween-20 (0.05%), bound antibodies were detected using anti-mouse-HRP using TMB substrate and colorimetric detection (Glomax Explorer, Promega).

### Flow cytometry
L-TGF-β1_lasso1/GARP or L-TGF-β1_lasso 3/GARP expressing TMLC cells were stained with anti-GARP clone 7B11 or anti-TGF-β1 clone 28G11 (Biolegend). Antibody binding was detected using goat-anti-mouse APC conjugate (Biolegend Poly-4053). eGFP was used as a surrogate marker for L-TGF-β1 expression. Analysis of expression was determined via flow cytometry using an LSR II (BD biosciences).

### TGF-β activation assays
We used stable transfection of Mink lung TGF-β reporter cells TMLC[30] (Gift from John Munger, NYU medical center, NYC, NY) with a vector containing either a WT human TGF-β1 IRES GFP, or a human TGF-β1 (R249A) IRES GFP cassette with puromycin resistance to obtain TMLC cells expressing WT L-TGF-β either capable of dissociating into LAP and mature TGF-β or not, due to the R249A mutation that normally allows furin cleavage of LAP from mature TGF-β[3]. Stable transfection of these lines with a HA-GARP construct (blastacidin resistance cassette) resulted in L-TGF-β surface expression, which was enhanced by selection and sorting to achieve high surface expression of TGF-β1/GARP or TGF-β1 (R249A)/GARP, as measured by anti-HA (clone 5E11D8, GenScript, Piscataway, NJ) or anti-LAP (R&D Systems, AF426).

The αvβ6 ectodomain or controls, αvβ3 (R&D Systems), BSA (Sigma-Aldrich), or anti-LAP (R&D AF426, 1 μg/ml) were immobilized on wells at varying concentrations, and non-specific binding sites were blocked by BSA, as described in ref. [4]. αvβ6-ND or the αvβ6 ectodomain were applied to wells containing TMLC cells expressing WT L-TGF-β1/GARP, and TGF-β activation was estimated by luciferase activity, as previously described[5].

### Reporting summary
Further information on research design is available in the Nature Portfolio Reporting Summary linked to this article.

## Data availability
The cryo-EM reconstructions and atomic models of the mutant αvβ8/L-TGF-β/GARP/28G11 is deposited to EMDB under the accession code: EMDB-47130 [https://www.ebi.ac.uk/pdbe/entry/emdb/EMD-47130]. Source data are provided with this paper.

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

## Acknowledgements
Equipment at the UCSF cryo-EM facility was supported by NIH grants S10OD020054, S10OD021741, and S10OD025881 and is managed by D. Bulkley and G. Gilbert. We thank M. Harrington and J. Li for computational support. We thank John Munger (NYU Medical Center, NYC, NY) for the TMLC cell line used to make the GARP/L-TGF-β cell line. This work was partially supported by NIH R01HL134183 and R01HL165175 (S.L.N. and Y.C.). Y.C. is an investigator of the Howard Hughes Medical Institute. BioRender was used for some figure preparations.

## Author contributions
M.J., R.I.S., Y.C. and S.L.N. conceptualized the project. M.J., R.I.S., T.S., and S.L.N. designed and generated constructs, M.J. and L.W. purified protein samples, M.J. prepared cryo-EM samples, conducted quality assessment by mass photometry, and SDS-PAGE. M.J. performed data collection, performed cryo-EM data processing, model building, and structure analysis, J.L. performed the final map deconvolution, R.I.S. and S.L.N. conceived and generated the cell lines, and designed the cell assay, and R.I.S. and T.S. performed the TGF-β activation assays, M.J., R.I.S., Y.C. and S.L.N. wrote the manuscript.

## Competing interests
S.L.N. is on the scientific advisory board (SAB) of Corbus Pharmaceuticals, LLC. Y.C. is on the SABs of ShuiMu BioSciences Ltd. and Pamplona Therapeutics. The remaining authors declare no competing interests.
