## [Transparent Peer Review file · Nature Communications]

Latent-TGF- β has a domain swapped architecture

Corresponding Author: Dr Stephen Nishimura

Version 0:

Reviewer comments:

Reviewer #1

(Remarks to the Author)

The manuscript by Jin et al reports cryoEM analyses of the latent TGF-b1 structure. Using a nifty and clever experimental approach, the authors unequivocally demonstrate that the LAP homodimer itself has a domain-swapped architecture, namely that the arm domain of a LAP monomer on one side is linked to the straitjacket domain on the other side (contralateral architecture). This is the second instance of domain-swapping reported within the latent TGF-b1 molecule. A first domain-swap, reported by Springer and colleagues in 2018, is located at the level of the linker between the LAP arm and the mature TGF-b1 domains (ref14 cited in the manuscript, by Zhao et al, J Biol Chem 293, 1579-1589).

I have previously reviewed 2 versions of the manuscript, which was submitted to NSMB before its recent transfer to NCOMM. Many of my requests for clarifications or improvements have been addressed by the authors in the current version of the manuscript and/or in the point-by-point replies provided by the authors.

I have understood that the main aim of this manuscript is the demonstration of the contralateral architecture of LAP homodimer. Indeed, from a technical point of view, the current manuscript demonstrates rather convincingly that this the case under the employed experimental conditions and the way recombinant LAP was prepared for the different experimental undertakings.

However, the obvious implication for this finding is whether it might bear functional importance with respect to TGF-b1 activation. This reviewer had suggested to clarify the biological significance of the authors' observation. In the absence of this context, this work remains technically elegant and sound but lacks the publication elements one might expect for papers at a publication forum such as Nature Communications. Unfortunately, convincing arguments to this effect are absent in the point-by-point reply or in the manuscript:

(1) How the demonstration of the LAP contralateral architecture supports the suggested model of activation by entropy redistribution is still not clear. In other words, the authors do not explain why the alternative ipsilateral architecture would not be compatible with activation by entropy redistribution.

(2) The question of the steric hindrance to binding by the 2 TGF-b receptor chains (TGFbR1 and TGFbR2) within the latent TGF-b1 molecule in the context of activation by entropy redistribution has not been addressed by the authors, neither in this manuscript, nor in their two previous reports (Campbell et al, Cell 2020; Jin et al, Cell 2024). In the manuscript currently under revision, new Figures 1a and 1b illustrate the very well established steric hindrance observed for TGFbR1 and TGFbR2 binding when TGF-b1 is latent, but no structural modelling is provided to show how this steric hindrance could be overcome by entropy redistribution upon activation by integrin α V β 8 or 6. In Campbell et al (2020), Figure 7 models TGFbR2 (but not TGFbR1) binding to one side of a latent TGF-b1 dimer presented by GARP. In Jin et al (2024), Figure 6E shows that TGFbR2-Fc does not bind to latent TGF-b1/GARP complex bound to plastic-immobilized integrin α V β 8, whereas TGFbR2-Fc does bind to latent TGF-b3/GARP complex. Thus, neither whether nor how "partially exposed" mature TGF-b1 could bind to the functional TGFb receptor (R1 + R2) and signal is demonstrated or modelled. Furthermore, mature TGF-b1 undergoes drastic conformational changes compared to the equivalent segment encapsulated in LAP, leading to the restructuring and exposure of critical structural elements to mediate receptor binding. This is readily demonstrable by the many structures of TGFb1 in the context of its standalone mature form (e.g. pdb entry 1KLA), its pro-form (e.g. pdb entry 5VQP), its encapsulation in LAP (e.g. from the authors' prior work, pdb entry 8VSD) etc.

Therefore, collectively, TGF-b1 activation via entropy redistribution presents two major fundamental challenges. None of them are convincingly addressed in the revised manuscript to enable linkage of the proposed domain-swapping and ensuing contralateral structure to function.

(3) Upon further consultation of the plethora of structural studies on LAP this reviewer remains baffled about certain inferences by the authors bearing relevance for the functional context of the claims in the manuscript. For instance, the asymmetry observed in the latent TGF-b1/GARP complex (described in Liénart et al, Science 2018 and cited as Ref2 in the submitted manuscript) is incorrectly described and interpreted in the manuscript introduction and in the point-by-point reply to reviewer comments, respectively. In the introduction, the authors state that “GARP asymmetrically anchors L-TGF-b on immune cell surfaces”. However, Liénart et al (2018) report that latent TGF-b1 is asymmetrically anchored to GARP (not to the cell membrane). In the point-by-point reply, the authors use this asymmetry argument to support the significance of the contralateral architecture and the proposed model of entropic activation as important to understand autocrine activity of TGF-b1 : “Considering that L-TGF-b1/GARP is asymmetrically anchored in plasma membrane, contralaterally exposed mature TGF-b1 would be located closer to its receptors nearby.” This assumption is not correct, as the asymmetry is not with regards to the plasma membrane, and hence cannot support the link that is made between activation by entropy redistribution and autocrine signaling.

(4) Another argument used by the authors in the point-by-point reply to explain the potential biological significance of the contralateral architecture is the following: “Beyond the activation mechanism, defining correct domain architecture also has clinical implications. At least two stabilizing antibodies to L-TGF-b, are currently in clinical trials. Both bind only one side of the L-TGF-b1 ring, due to steric hinderance or the presence of GARP. The domain-swapped architecture together with our new conceptual model of TGF-b activation may explain why these antibodies only incompletely block integrin-mediated TGF-b activation.” Based on the published literature, the authors’ comment is rather confusing. The authors should specify which two antibodies they are referring to if one is to evaluate the validity of their argument.

Reviewer #1 (Remarks to the Author):

The manuscript by Jin et al reports cryoEM analyses of the latent TGF- β 1 structure. Using a nifty and clever experimental approach, the authors unequivocally demonstrate that the LAP homodimer itself has a domain-swapped architecture, namely that the arm domain of a LAP monomer on one side is linked to the straitjacket domain on the other side (contralateral architecture).

We appreciate this conclusion, which clears away any concerns over technical issues related to experiments that we conducted. In this rebuttal, we focus on the implications of our findings, and other questions that are unrelated to the findings of this manuscript.

This is the second instance of domain-swapping reported within the latent TGF- β 1 molecule. A first domain-swap, reported by Springer and colleagues in 2018, is located at the level of the linker between the LAP arm and the mature TGF- β 1 domains (ref14 cited in the manuscript, by Zhao et al, J Biol Chem 293, 1579-1589).

Indeed, the Springer lab biochemically clarified one of the two unresolved linkers in the crystal structure of L-TGF- β between LAP and mature TGF- β 1, but not the linker between the arm and straitjacket domains (JBC, 2018, ref 14). That work was closely related to an unresolved question in an earlier study also from the Springer lab (Nature 2017, ref 10), in which the arbitrarily assigned non-swapped ipsilateral model was used to perform MD simulation of the α v β 6/L-TGF- β 1 complex. The findings were that ipsilateral straitjacket domain was the first to be deformed by the tensile force applied from the integrin β 6 subunit through the ipsilateral arm domain, consistent with the intuitive idea of force being most effectively transduced through domains that are physically linked. However, recognizing the caveat of arbitrarily assigned domain architecture, the authors also stated: *“As the effect of force on domains or multi-domain assemblies is highly dependent on the direction of the force vector, physiological orientation between integrins and their macromolecular ligands is necessary to understand the biological consequences of force transmission”*. This statement explains their follow up studies to clarify the domain architecture (JBC, 2018, ref 14). Together, these are an attestation that the domain architecture of L-TGF- β is of high interest and significant biologic implication to the integrin-mediated TGF- β activation mechanism. Our study in this manuscript has now resolved this key structural feature of L-TGF- β .

I have previously reviewed 2 versions of the manuscript, which was submitted to NSMB before its recent transfer to NCOMM. Many of my requests for clarifications or improvements have been addressed by the authors in the current version of the manuscript and/or in the point-by-point replies provided by the authors.

We thank the Reviewer for her/his extensive efforts reading multiple versions of this manuscript and providing valuable insights and comments helping us to improve the manuscript. We appreciate the acknowledgment by the Reviewer that we have addressed many of her/his comments in the current revision. In the following, we provide our further responses, addressing additional comments raised.

I have understood that the main aim of this manuscript is the demonstration of the contralateral architecture of LAP homodimer. Indeed, from a technical point of view, the current manuscript demonstrates rather convincingly that this is the case under the employed experimental conditions and the way recombinant LAP was prepared for the different experimental undertakings.

We appreciate that the Reviewer agrees that we demonstrated convincingly that L-TGF- β has a domain swapped/contralateral architecture.

However, the obvious implication for this finding is whether it might bear functional importance with respect to TGF- β 1 activation. This reviewer had suggested to clarify the biological significance of the authors' observation. In the absence of this context, this work remains technically elegant and sound but lacks the publication elements one might expect for papers at a publication forum such as Nature Communications. Unfortunately, convincing arguments to this effect are absent in the point-by-point reply or in the manuscript:

Our response above has already addressed the question of the biological significance of the correct domain architecture of L-TGF- β . The following comments from Reviewer 1 went beyond the biological significance of L-TGF- β architecture rather focusing on questions outside of the subject of this manuscript, namely, results reported by us that TGF- β activation does not require being released from latent complex (Cell, 2020, ref 4) and our proposed model that dynamic allostery driven by entropy redistribution activates TGF- β for autocrine signaling (Cell, 2024, ref 5). As we stated in earlier revisions, and will re-emphasize here again, the purpose of this manuscript is NOT to support the entropy redistribution model, but to report the correct architecture of L-TGF- β , which is broadly important to mechanistic understanding of integrin mediated TGF- β activation, either by tensile force generated by avb6 or dynamic allostery driven by entropy redistribution by avb8, or by any other mechanism.

Nonetheless, we further provide our response below:

(1) How the demonstration of the LAP contralateral architecture supports the suggested model of activation by entropy redistribution is still not clear. In other words, the authors do not explain why the alternative ipsilateral architecture would not be compatible with activation by entropy redistribution.

Again, we would like to reiterate that the primary purpose of this manuscript is NOT to support the entropy redistribution model we proposed for avb8 mediated L-TGF- β activation proposed in our previous study (Cell, 2024). In that study, we observed the reduction of flexibility at the integrin binding site coincided with the increase of flexibility of contralateral lasso loop and interpret this observation as the transduction of flexibility from the arm domain to the contralateral straitjacket/lasso induced by avb8 binding. This is further conceptualized as conformational entropy redistribution hypothesis. The next question was through which path entropy is redistributed from arm domain to the contralateral lasso. Several factors promoted us to assign a contralateral domain architecture. 1) We observed a weak density in cryo-EM map that physically links integrin binding site to the contralateral lasso loop, 2) AlphaFold prediction gave two possible domain architectures, domain swapped contralateral architecture and domain non-swapped ipsilateral architecture, and 3) intuitively thinking and in reference to MD simulation conducted by

the Springer lab (Nature 2017, ref 10) that force is transduced through one monomer, we hypothesized that entropy redistribution might be more efficient through the same monomer. Together, these data promoted us to model L-TGF-b as the domain swapped architecture, in which flexibility is transduced from integrin binding site to the lasso loop of the same monomer. However, we did not have definitive experimental data to support this architecture. In this manuscript, we now present the clear and convincing data to demonstrate the correct architecture is indeed domain swapped, as we hypothesized in our previous study. In this sense, the finding is consistent with our previous model.

With respect, we are unclear as how studying the ipsilateral domain architecture would clarify the activation mechanism since this architecture does not exist. Instead, we emphasize that knowing the correct domain architecture is critical in designing future experiments to further test the tensile force and entropy redistribution hypotheses, and if they are correct, partially correct or incorrect. But these future studies are beyond the scope of the current manuscript.

As stated above, we have refrained from claiming that the correct architecture supports the entropy redistribution model, but rather discuss that the domain architecture when placed in the context of the entropy redistribution or tensile force hypothesis provides the structural pathway for either activation process.

(2) The question of the steric hindrance to binding by the 2 TGF-b receptor chains (TGFbR1 and TGFbR2) within the latent TGF-b1 molecule in the context of activation by entropy redistribution has not been addressed by the authors, neither in this manuscript, nor in their two previous reports (Campbell et al, Cell 2020; Jin et al, Cell 2024).

In the manuscript currently under revision, new Figures 1a and 1b illustrate the very well established steric hindrance observed for TGFbR1 and TGFbR2 binding when TGF-b1 is latent, but no structural modelling is provided to show how this steric hindrance could be overcome by entropy redistribution upon activation by integrin α v β 8 or 6. In Campbell et al (2020), Figure 7 models TGFbR2 (but not TGFbR1) binding to one side of a latent TGF-b1 dimer presented by GARP. In Jin et al (2024), Figure 6E shows that TGFbR2-Fc does not bind to latent TGF-b1/GARP complex bound to plastic-immobilized integrin α v β 8, whereas TGFbR2-Fc does bind to latent TGF-b3/GARP complex. Thus, neither whether nor how “partially exposed” mature TGF-b1 could bind to the functional TGFb receptor (R1 + R2) and signal is demonstrated or modelled.

First, as mentioned above, the purpose of this manuscript is to report the correct architecture of L-TGF-b, which by itself is critically important for understanding its activation mechanism. Thus, with respect, we would like to point out that using this argument against publication of our study is unreasonable.

Secondly, we respectfully disagree with Reviewer 1 that we didn't demonstrate association of TGF-bR2 with partially exposed TGF-b1. In our two early papers (Cell, 2020 and 2024, ref 4, 5), we presented three important pieces of data which together demonstrated that α v β 8 binding induces TGF-b activation without release and therefore that TGF-b receptors can overcome steric hindrance and interact with TGF-b when still associated with the latent complex: 1) We demonstrated structurally in both papers that the L-TGF-b ring becomes dramatically flexible upon

avb8 binding. Such flexibility is the result of drastic **deformation** of latent ring. If such deformation is sufficient to expose mature TGF-b and relieve steric hinderance of the lasso loop is the next question. 2) We addressed this question by demonstrating that TGF-b signaling is activated upon avb8 binding without releasing mature TGF-b (with a furin cleavage site mutation), in vitro (2020, ref 4) and in vivo (2024, ref 5). The in vivo evidence that TGF-b without release can still generate TGF-b signaling is a highly rigorous test of the biologic relevance of the hypothetical model that mature TGF-b can be exposed to its receptors without being released. 3) As pointed out by the reviewer, we did demonstrate biochemically that TGFbR2 binds to avb8-bound L-TGF-b3/GARP better than to L-TGF-b1/GARP, the later which only showed a small signal over baseline. These data provide biochemical evidence that TGF-bR2 can overcome steric hinderance to bind to TGF-b when still physically associated with LAP (Figure 6E of 2024). That TGF-bR2 binding to L-TGF-b1/GARP is more difficult to detect in vitro is consistent with our structural data showing that L-TGF-b1 is much more stable than L-TGF-b3. Together, these data support a model that explains why avb8 binding deforms L-TGF-b and activates TGF-b autocrine signaling, without release. Currently, there is no other hypothesis that accounts for our collective data.

Indeed, we have thus far been unable to capture the exact structure showing how deformation of the L-TGF-b ring exposes mature TGF-b and allows binding to its receptors. Giving the dynamic nature of the deformed L-TGF-b, determining its structure represents an intrinsic limitation of single particle cryo-EM (or crystallography in this regard). This same limitation applies to L-TGF-b deformed by tensile force and explains why there is arguably even less structural evidence supporting integrin-mediated TGF-b release, which historically was the first and therefore the most cited mechanism of integrin-mediated TGF-b activation in the literature. Without capturing the exact structure of receptors bound to mature TGF-b not released from latent ring does not mean such binding does not exist. A scientifically reasonable approach is to establish structure-based models/hypotheses, which then allow design of further experiments to directly and indirectly test the hypothesis, or to generate new hypotheses based on the all available data.

Furthermore, mature TGF-b1 undergoes drastic conformational changes compared to the equivalent segment encapsulated in LAP, leading to the restructuring and exposure of critical structural elements to mediate receptor binding. This is readily demonstratable by the many structures of TGFb1 in the context of its standalone mature form (e.g. pdb entry 1KLA), its pro-form (e.g. pdb entry 5VQP), its encapsulation in LAP (e.g. from the authors' prior work, pdb entry 8VSD) etc.

We agree with the Reviewer that mature TGF-b1, and similarly mature TGF-b3 undergo conformational changes upon TGF-bR binding. As the reviewer pointed out, we have already shown that mature TGF-b3 can bind to TGF-bR2 when still associated with its latent complex, thus providing evidence that whatever conformational change occurs upon TGF-bR2 binding can be accommodated when TGF-b3 is still associated with LAP. Our cell based and in vivo data suggest that the same applies to TGF-b1. Not being able to obtain a structure of an intermediate complex, does not mean it does not exist, especially if there is definitive biological evidence that it does exist.

Therefore, collectively, TGF-b1 activation via entropy redistribution presents two major fundamental challenges. None of them are convincingly addressed in the revised manuscript to

enable linkage of the proposed domain-swapping and ensuing contralateral structure to function.

This comment suggests that the main question of the Reviewer is with our previously published work, specifically the data that TGF- β can be activated without being released from the latent LAP and our hypothesis that entropy redistribution drives avb8 mediated TGF- β activation. The comment does not specifically address the quality nor the interpretation of the data under consideration, only its context with our previously published work. The title of the subject at hand is “TGF- β has a domain swapped architecture” and the reviewer has acknowledged that we have convincingly demonstrated this. Nonetheless, we will address this question in the following:

Up to now, there are only two mechanistic models concerning integrin-mediated TGF- β activation, the tensile force and entropy redistribution, both of which are hypothetical. The domain swapped architecture accommodates both hypotheses. For unclear reasons, the entropy redistribution hypothesis is the main point of contention from Reviewer 1, even though the tensile force hypothesis has less structural and in vivo evidence to support it than the entropy redistribution hypothesis.

We are perplexed as to why the absence of a demonstration of biochemical association of L-TGF- β 1 with TGF- β R2, as well as a structure, invalidate the significance of avb8 binding induces L-TGF- β deformation, without tensile force. As of today, this hypothetical model remains to be the only model that can explain all experimental data. Clearly, much remains to be learned about integrin-mediated TGF- β activation, and correct assignment of the domain architecture can only facilitate these studies. We never claimed definitively this is the correct model. Instead, as we stated in the “Limitation of Study” of 2024 Cell, it is a simplified and conceptual model. More work is needed to improve, modify it or even to replace it with a new hypothesis. The result presented in this current manuscript was NOT intended to provide additional support to the entropy redistribution model, but rather to resolve a key question that is critical to any mechanistic models proposed to explain integrin mediated L-TGF- β activation, tensile force model, entropy redistribution model, combination of the two, or any other new model to be proposed. Thus, this should not be a factor against publication of our current study.

(3) Upon further consultation of the plethora of structural studies on LAP this reviewer remains baffled about certain inferences by the authors bearing relevance for the functional context of the claims in the manuscript. For instance, the asymmetry observed in the latent TGF- β 1/GARP complex (described in Liénart et al, Science 2018 and cited as Ref2 in the submitted manuscript) is incorrectly described and interpreted in the manuscript introduction and in the point-by-point reply to reviewer comments, respectively. In the introduction, the authors state that “GARP asymmetrically anchors L-TGF- β on immune cell surfaces”. However, Liénart et al (2018) report that latent TGF- β 1 is asymmetrically anchored to GARP (not to the cell membrane). In the point-by-point reply, the authors use this asymmetry argument to support the significance of the contralateral architecture and the proposed model of entropic activation as important to understand autocrine activity of TGF- β 1 : “Considering that L-TGF- β 1/GARP is asymmetrically anchored in plasma membrane, contralaterally exposed mature TGF- β 1 would be located closer to its receptors nearby.” This assumption is not correct, as the asymmetry is not with regards to

the plasma membrane, and hence cannot support the link that is made between activation by entropy redistribution and autocrine signaling.

We agree that we can be more exact in the point we are making about asymmetry. Since L-TGF- β is asymmetrically anchored to GARP each straitjacket/lasso domain of L-TGF- β will be asymmetric relative to GARP. This asymmetry of the straitjacket/lasso domains will be maintained relative to GARP when anchored to the cell membrane. We have now revised this section in the manuscript to be clearer in our point.

(4) Another argument used by the authors in the point-by-point reply to explain the potential biological significance of the contralateral architecture is the following: "Beyond the activation mechanism, defining correct domain architecture also has clinical implications. At least two stabilizing antibodies to L-TGF- β , are currently in clinical trials. Both bind only one side of the L-TGF- β 1 ring, due to steric hinderance or the presence of GARP. The domain-swapped architecture together with our new conceptual model of TGF- β activation may explain why these antibodies only incompletely block integrin-mediated TGF- β activation." Based on the published literature, the authors' comment is rather confusing. The authors should specify which two antibodies they are referring to if one is to evaluate the validity of their argument.

The antibodies are SRK181 and MHG8. The data from the original publication of SRK181 shows that SRK blocks maximally ~75% of avb8-mediated cell surface L-TGF- β /GARP activation (PMID: 32213632). The functional data on MHG8 is from our published data which shows that MHG8 only weakly inhibits avb8-mediated cell surface TGF- β /GARP activation (Seed, et al, 2020). Structurally, MHG8 binds asymmetrically to one side of the L-TGF- β /GARP complex (PMID: 30361387). We have the structure of L-TGF- β /GARP/SRK181 showing it binds to L-TGF- β /GARP only in one orientation and have additional data showing that it also blocks avb8-mediated TGF- β /GARP activation partially. This work is part of a manuscript in preparation and is not yet published.